# A Review of Hand Function Rehabilitation Systems Based on Hand Motion Recognition Devices and Artificial Intelligence

**DOI:** 10.3390/brainsci12081079

**Published:** 2022-08-15

**Authors:** Yuexing Gu, Yuanjing Xu, Yuling Shen, Hanyu Huang, Tongyou Liu, Lei Jin, Hang Ren, Jinwu Wang

**Affiliations:** 1School of Biomedical Engineering, Shanghai Jiao Tong University, Shanghai 200030, China; 2Shanghai Key Laboratory of Orthopaedic Implants, Department of Orthopaedic Surgery, The Ninth People’s Hospital Affiliated to School of Medicine of Shanghai Jiao Tong University, Shanghai 200011, China; 3College of Science, Xi’an Jiaotong-Liverpool University, Suzhou 215028, China; 4Department of Rehabilitation Medicine, The Ninth People’s Hospital Affiliated to School of Medicine of Shanghai Jiao Tong University, Shanghai 200011, China; 5School of Health Science and Engineering, University of Shanghai for Science and Technology, Shanghai 200093, China

**Keywords:** hand function rehabilitation, hand rehabilitation robot, computer vision technology, wearable devices, sensors, artificial intelligence

## Abstract

The incidence of stroke and the burden on health care and society are expected to increase significantly in the coming years, due to the increasing aging of the population. Various sensory, motor, cognitive and psychological disorders may remain in the patient after survival from a stroke. In hemiplegic patients with movement disorders, the impairment of upper limb function, especially hand function, dramatically limits the ability of patients to perform activities of daily living (ADL). Therefore, one of the essential goals of post-stroke rehabilitation is to restore hand function. The recovery of motor function is achieved chiefly through compensatory strategies, such as hand rehabilitation robots, which have been available since the end of the last century. This paper reviews the current research status of hand function rehabilitation devices based on various types of hand motion recognition technologies and analyzes their advantages and disadvantages, reviews the application of artificial intelligence in hand rehabilitation robots, and summarizes the current research limitations and discusses future research directions.

## 1. Introduction

A stroke is one of the most common causes of adult labor loss. It significantly affects people’s quality of life. A patient may endure certain movement disorders after a stroke, such as paralysis of the face, arm, and leg on one side of the body. This is known as hemiplegia [1]. Reduced motor function of the upper limbs, especially the hands, essentially limits the ability of the patients to perform activities of daily living (ADL). The hand, a distal part of the body, is the most challenging part of the upper limb to recover [2]. Accordingly, the degree of rehabilitation of hand function can also be used to measure the rehabilitation of the upper limbs from movement disorders. In recent years, many therapeutic methods for upper limb movement recovery after strokes have been developed, among which rehabilitation robots are considered to be an efficient rehabilitation training method [3], which can not only help patients recover the motor function of their limbs, but also significantly reduce the burden of rehabilitation therapists [4].

Effective hand function rehabilitation training based on rehabilitation systems should, at least, meet the following three principles [5,6]: first, the rehabilitation system can ensure the training motivation of patients and help patients stick with the training. In addition, its training program should be customized according to patients’ conditions. Furthermore, the rehabilitation system needs to be able to objectively assess the patient’s hand function and training outcomes on a regular basis. Therefore, acquiring and recognizing hand postures is crucial in the rehabilitation and evaluation of hand function. It has the following applications. First, it can be applied to the active training mode of rehabilitation robots. Specifically, the active motion intention of patients can be stimulated, extracted, and utilized through detection technology and modern control technology. This can not only promote the recovery of patients’ motor perception, but also help patients reshape the central nervous system circuit. Secondly, gesture recognition can also be applied in the human–computer interaction module of the telerehabilitation system and the evaluation system. The main methods of gesture acquisition can be divided into computer vision technology and sensor-based wearable devices according to the different input devices. The former uses external devices for vision capture and analysis, which poses no interference to patients. The latter requires wearing some sensor devices and recognizes fewer gestures, but is beneficial for portability.

In the development of hand rehabilitation robots, software systems are equally important in the design and production of hardware devices. With the development of artificial intelligence, big data, cloud computing, and 5G technology, the requirements for the software system of hand rehabilitation robots have also increased. It should not only fulfill the basic hardware control function and the human–machine interaction function, but also be more intelligent, diverse, and personalized [7]. Artificial intelligence can be applied to many modules of hand rehabilitation robots, which can not only expand the functions of hand rehabilitation robots, improve the accuracy, effectiveness, and wisdom of the devices, but also reduce the pressure on medical resources and improve the comfort and fun of patients during rehabilitation [8].

The outline of this paper is as follows: Section 2 reviews the developments in the hardware of the hand function rehabilitation systems, which mainly includes the gesture recognition devices applied based on computer vision technology and wearable sensors. Section 3 reviews the developments in the software of the hand function rehabilitation systems, including application of artificial intelligence in the seven modules of the hand function rehabilitation robot, although some of them are still in the research stage and whose actual rehabilitation effects are yet to be verified. Section 4 lists the existing problems and limitations during the current phase and discusses the potential directions for future study.

## 2. Hand Function Rehabilitation System Based on Gesture Recognition

### 2.1. Hand Function Rehabilitation System Based on Computer Vision Technology

Hand gesture recognition based on computer vision technology mainly uses external devices to collect image data of gestures, such as cameras, and then processes the images with vision techniques, such as deep convolutional networks, to complete recognition and classification. This method is non-invasive, does not require wearing extra equipment, the user is not easily fatigued, and the calibration procedure is simple and convenient to use. It can be applied to fine hand rehabilitation training and hand function assessment tasks. However, this method also has some shortcomings. First, it has high requirements for the external environment. Second, the recognition speed and accuracy of it are relatively low. Third, sometimes, it needs to affix specific markers. The devices that collect gestures usually include color cameras, depth cameras, etc. These hand gesture recognition techniques based on computer vision have been applied to hand rehabilitation robots and independent hand function rehabilitation game systems.

#### 2.1.1. Hand Function Rehabilitation System Based on the Virtual Environment

First of all, some cameras and computer vision techniques can be used to locate the hand’s position and recognize some basic hand gestures, which will provide patients with some more exciting games through augmented reality and virtual reality to enhance their interest in training. Studies have shown that video games are beneficial for improving cognitive dysfunction in patients [9]. Other studies indicated that virtual environment training is significantly more effective than traditional training [10,11,12,13,14,15]. A virtual environment also provides a safe and customizable training system that can be changed according to users’ interests. It can also monitor users’ actions to analyze their performance during training [16].

For example, Hondori et al. [17] developed a low-cost augmented reality system for hand rehabilitation training assistance and progress assessment, which provides task instructions through virtual objects projected on the desktop, and uses a web camera to collect and track the color markers of hand recognition to identify patient actions and task completion. This device can allow patients to train in the hospital or at home and can be remotely accessed and controlled by therapists. However, some additional markers need to be worn, which is not very convenient, and the hand positioning recognition ability also needs to be improved.

In addition to webcams, similar hand modeling, tracking, and recognition functions can be accomplished using depth cameras and color cameras without markers. The primary devices currently used are Leap Motion, Microsoft Kinect V1 and V2, etc.

The Leap Motion uses infrared LED and a gray-scale camera, which is cheaper and can handle hand models only. Wang et al. [18] demonstrated that Leap Motion-based virtual reality training could promote cortical reorganization and may aid in the recovery of upper extremity motor function in patients with subacute stroke. However, its effects on severe patients are unknown. Alimanova et al. [19] developed a set of hand rehabilitation games using Leap Motion controllers to help patients relax and train their hand muscles by performing different virtual reality tasks related to daily living activities, such as picking up objects, moving household objects, matching color blocks, throwing garbage, holding objects, etc. These games can also motivate patients by making the rehabilitation process more exciting and effective, and they can help patients relieve muscle tension and restore hand function.

Mukai et al. [20] developed the hand rehabilitation robot “ReRoH”. It comprises flexible pneumatic gloves, ERB, an electrical stimulator, Leap Motion non-contact sensor, and game controller. The Leap Motion has an infrared transmitter and receiver that can identify hand coordinates and movements, and the game controller can display an image of the hand on a display in real-time. With rehabilitation games, the ability to grasp and stretch the hand can be trained, and the motor function of the fingers and hands can also be assessed.

The Kinect uses depth cameras, infrared emitters, color sensors, infrared depth sensors, turnover motors, RGB cameras, etc., all of which are needed to generate a full-body or half-body model, resulting in the latter being less efficient and accurate in operation than the former.

For example, Cipresso et al. [21] developed a virtual reality system for rehabilitation by using Kinect V1. It combines rehabilitation technology with virtual environment recognition. It can track patients’ hand motions through Kinect’s depth images and color space model, and provides real-time feedback. Through the virtual reality of daily life activities with highly realistic visual effects, patients’ motivation to participate in treatment is stimulated, and their ADL ability is better trained and evaluated. Likewise, Wang et al. [22] developed a Kinect-based rehabilitation assistance system consisting of the rehabilitation training subsystem, and rehabilitation evaluation subsystem that performed similar functions.

However, these systems lack tactile feedback, game angle, and time-based safety constraints based on the patient’s motor ability. Saini et al. [23] solved this problem by designing a “watchdog” to ensure the safety of patients’ hand motion angle and time during the game. Additionally, the game framework of the stroke rehabilitation program designed by them only uses Kinect V1 equipment and a single camera to capture human motion, without the need to affix markers.

Avola et al. [24] proposed an interactive and low-cost full-body rehabilitation framework for generating 3D immersive serious games. The hardware part includes Kinect V2 and a head-mounted display (HMD), and acquires depth information in the way of time of flight (TOF), which is different from Kinect V1 using light coding. The software part uses gated recurrent unit recurrent neural networks (GRURNN). The framework’s natural user interface (NUIs) can be used for hand modeling, tracking the movement of hand joints and fingers through an infrared stereo camera, and then performing rehabilitation training through a customizable interactive virtual environment. Experiments showed that this system can restore patients’ hand function. However, it only has a few hand games to choose from, and HMD may cause vertigo in some patients.

#### 2.1.2. Hand Rehabilitation Robot Based on Computer Vision Technology

In addition to the above applications, computer vision technology can also be used simply for gesture recognition. The hand rehabilitation robot can be controlled by recognized gestures to carry out the corresponding motion. It can also identify the current motion state of the rehabilitation robot and provide feedback to the system for further adjustment of the robot or evaluation of the training situation.

Cordella et al. [25] developed a set of camera-based calibration programs for the bending sensor of the commercial exoskeleton hand rehabilitation robot glove “Gloreha Sininfonia,” which uses 8 photoelectric cameras to locate and reconstruct the angles of 18 reflective markers on the glove, and then connects them with the voltage of the glove bending sensor to form closed-loop control, in order to calibrate the angle of the exoskeleton better. The system can also assess a patient’s hand function or measure the improvement of hand motion by comparing their range of motion (ROM) before and after treatment.

Farulla et al. [26] proposed a hand exoskeleton system that can be used for master-slave control of telerehabilitation. It uses an RGB-D camera to locate the position of the therapist’s hand joint in real-time through the VPE algorithm and remotely transmits the position to the exoskeleton. In this way, the movement of the index finger and thumb of the patient’s hand can be controlled. The grip force sensor on the exoskeleton can record the interaction force and feed it back to the therapist for real-time quantitative assessment and adjustment.

Nam et al. [27] developed an exoskeleton that can remotely train the ability of hand grasping and forearm pronation and supination, and also used camera-based computer vision technology to identify the user’s intention. However, unlike other systems, this device is equipped with a Microsoft LifeCam Studio camera on the exoskeleton. The patient uses the residual force of the proximal upper limb to move the robot to the target position and identify the target object. Firstly, the recognition algorithm identifies and locates the target through the real-time image collected by the camera. Then, the control system controls the aiming and grasping of the exoskeleton. Finally, the system realizes the motion-vision-sensation closed-loop feedback. To prevent hand injury caused by excessive grasping movements of the robot, the exoskeleton of the hand is placed on the palm side.

There are also many related types of research and achievements in China. For example, Hefei University of technology [28,29,30,31] developed a rehabilitation robot system. When the patient completes the specified gesture in front of the camera on the mechanical arm, the processor will display it on the screen after recognition through the internal algorithm. Then, the system can analyze, evaluate, and record the patient’s rehabilitation. It can also cooperate with the virtual reality system for more exciting training to improve the patient’s cognitive ability and hand function and adjust the training scheme as needed. When the camera cannot capture the complete hand image, the system can automatically identify and locate the position of the patient’s hand, adjust the manipulator to the position where the complete gesture can be captured, and use the exoskeleton to compensate for the hand motion.

Nanchang University [32] developed a near-synchronous hand rehabilitation robot. It can collect the motion posture information of the unaffected hand in real-time through Leap Motion, decode and transmit the data through a genetic algorithm and neural network, and then control the exoskeleton to complete the corresponding action and drive the patient’s hands for training. The system has a fast response time and high accuracy in gesture recognition and control, but it can only train the thumb, index finger, and middle finger.

Liu Hongmei et al. [33] from Shanghai Normal University developed a flexible hand rehabilitation robot by using the Vicon system, which has eight infrared cameras. They can capture the three-dimensional motion of the hand, obtain the joint angle, and recognize the hand motion to identify the relationship between the extension of the Bowden line and the bending angle of the finger, so as to control the hand rehabilitation robot more accurately.

### 2.2. Hand Function Rehabilitation System Based on Wearable Devices

Gesture recognition based on wearable devices requires users to wear devices, such as gloves, rings, bracelets, wristbands, armbands, etc., and collect the motion data or physiological signal data of the user’s hand movements through the sensors on the devices for recognition. This method can recognize more gestures with small input data and high precision. It can identify hand motion and three-dimensional information in space in real-time. It is not easily disturbed by external interference and has good robustness, but it is usually expensive. Wearing equipment may interfere with the therapist’s motion, and the accuracy may be affected when the user is sweating. It is also easy to make the patient tired, and calibration is usually required before each use. According to the types of data collected, wearable devices can be divided into physiological signal sensors, kinematic signal sensors, optical signal sensors, etc. Multiple transmissions are often selected simultaneously as multi-mode fusion data inputs in actual use. Gestures recognized by wearable devices can provide control targets for hand function rehabilitation robots and complete hand function training in the active mode. They can also be used for the evaluation of a patient’s rehabilitation.

#### 2.2.1. Physiological Signal Sensor-Based Hand Function Rehabilitation Robot

The bioelectric signals that can be used for hand function rehabilitation robots usually include electroencephalogram (EEG), electro-oculogram (EoG), electromyography (EMG), etc. Generally, the acquisition system collects the required bioelectric signals from the patient’s body surface, processes and analyzes the signals to obtain the patient’s motion intention, which is equivalent to synchronously identifying the patient’s hand action, and then transmits the corresponding motion instructions to the hand rehabilitation robot to drive the patient’s hand to make related motions.

Wang Jing et al. [34] from Xi’an Jiaotong University developed the upper limb rehabilitation platform by using the commercialized product Myo gesture control armband to recognize the gesture intention of patients. Gestures are then mapped to a virtual environment, and patients’ hand grasping functions are trained through VR glasses and immersive games. The armband is composed of an inertial sensor unit, eight surface electromyography (sEMG) sensors, and a Bluetooth receiver, which is low-cost and portable. Liu Wei et al. [35] from Nanjing University of Aeronautics and Astronautics designed an underactuated hand rehabilitation robot, which also uses MYO gesture control armbands. It can recognize five gestures of the unaffected hand in real-time, and control the exoskeleton to drive the affected hand to complete the corresponding movements for training. Similarly, the low-cost exoskeleton “RobHand” developed by Casnal et al. [36] also completes the bilateral cooperative control of the rehabilitation robot by fixing multiple sEMG sensors. Li et al. [37] designed an exoskeleton that selects 16 muscles in both arms and hands for sEMG acquisition. Combined with the optimized algorithm, it can recognize seven gestures and control the exoskeleton. In addition, it can also estimate the strength of the patient when grasping the object by collecting sEMG signals, and control the rehabilitation robot to provide the required auxiliary force, such as the exoskeleton designed by Leonardis et al. [4]

Soekadar et al. [38] from Germany developed the hand function rehabilitation exoskeleton. This exoskeleton uses an innovative brain/neurocomputer interaction (BNCI) system that integrates EEG and EOG to better compensate for the decline in signal quality and fatigue sensitivity over time. This system can also recognize gestures better and control the exoskeleton. Similarly, Huo Yaopu et al. [39] from Southeast University designed a 3D-printed hand exoskeleton based on the motor imagination brain computer interface (MI-BCI), which also identifies the motion intention of patients by collecting their EEG, and then controls the hand exoskeleton to assist them in completing corresponding actions.

Zhang et al. [40] from Changsha Institute of Mechanical Engineering proposed the multi-mode human–computer interaction flexible rehabilitation robot, which uses three modes, EEG, EOG and EMG. Under the condition of meeting the requirements of classification speed, it improves the accuracy of the classification and dramatically improves the performance and applicability of the system, but also reduces the system’s response speed. Similarly, Xi’an Jiaotong University [41] designed the multi-mode human–computer interaction flexible manipulator, which also integrates the instructions of three modes, so that patients can choose their own multi-modal human machine interface (mHMI) mode.

Bioelectrical signals to control hand rehabilitation robots have considerable application prospects. This mode is more in line with the physiological function of the human body; that is, there are physiological signals first, and then muscle activities to produce hand movements. However, the balance and optimization between comfort, diversity, stability, accuracy, timeliness, and timeliness need to be further studied, and the safety of unsupervised training is also worthy of attention.

#### 2.2.2. Kinematics Sensor-Based Hand Function Rehabilitation Robot

The kinematic parameters of hand motion include fingers and joints’ coordinates, displacement, angular displacement, motion speed, angular velocity, acceleration, inertia, etc. The kinematic sensors used to measure these parameters in a hand rehabilitation robot mainly include flexible angle sensors, accelerometers, gyroscopes, pressure sensors, etc. Compared with rehabilitation robots based on physiological signals, most rehabilitation robots with integrated kinematic sensors can provide a signal of superior quality and better tolerance with regard to placement of electrodes. They can recognize finer gestures and complete finer-grained training tasks, which is very helpful for the treatment of patients in the later stages of stroke rehabilitation.

Most of the kinematic sensors of hand rehabilitation robots choose the angle sensors, because they can obtain the angle information of each finger joint more directly and then recognize more fine gestures, which can be used for hand function rehabilitation and evaluation. For example, Hong et al. [42] from the National University of Singapore designed a passive flexible mechanical glove. It uses the angle sensor placed on the finger part of the glove to collect the kinematic information of the patient’s healthy hand to judge the patient’s motion intention, and recognize the gesture. It then uses the pneumatic actuator made of silicone elastic material to generate pressure to drive the affected hand to move. Similarly, Rahman et al. [43] from Sydney University of Technology developed a 15-DOF aluminum hand exoskeleton that also adopts angle sensors and the bilateral cooperative mode.

When the patient has a certain degree of hand function, the tactile sensor and pressure sensor can also be used to identify the motion intention of the affected hand when grasping the object and judge the required force. For example, Nilsson et al. [44] from Sweden developed SEM gloves with tactile sensors on the fingertips and pressure sensors on the palm. After recognizing the patient’s motion intention of grasping and judging the vital force, the robot will provide the corresponding auxiliary power to complete the training.

Many devices simultaneously use angle and pressure sensors to obtain more comprehensive kinematic information. For example, Chen et al. [45] proposed a flexible and portable hand fine motor function rehabilitation robot, with ten flexible bending angle sensors at the joints and ten pressure sensors on the palm and fingertips of the unaffected hand’s glove. Through these sensors, the joint bending angle and clamping force of each finger can be collected in real-time. Then, 16 gestures and 6 task gestures of a single finger and multiple fingers motion can be recognized, and the affected glove can be controlled to complete the same action-driven training. There are also angle and pressure sensors on the gloves on the affected hand, which can provide feedback on the current parameters for closed-loop control and provide a basis for doctors to evaluate hand function and adjust the training program. Similarly, the rehabilitation robot designed by Rakhtala et al. [46] also uses angle and pressure sensors at the same position of gloves to identify the current hand motion state to achieve a closed-loop control system. The same is true of the pneumatic hand rehabilitation robot designed by Huazhong University of Science and Technology [47].

The angular information during a hand motion is not only the finger joint bending angle mentioned above, but also the opening angle and the degree of overlap between each finger. Sometimes, it is necessary to use an opening angle sensor to measure the degree of the spread between fingers, which can better identify different types of hand gestures and analyze the patient’s hand function rehabilitation more comprehensively. For example, Li Nan et al. [48] designed intelligent-assisted rehabilitation gloves, which use bending angle sensors, pressure sensors, and opening angle sensors at the same time.

For the recognition of dynamic hand motions, in addition to static motion information, some dynamic posture information, such as velocity and acceleration, and spatial information, such as relative coordinates, are required. At this time, accelerometers, gyroscopes, and other sensors are needed. These applications are often combined with inertial sensors or posture sensors in practice. For example, Zhengzhou University [49] designed the hand motion rehabilitation training and evaluation system. This system uses bending sensors and posture sensors simultaneously, which can obtain the motion state of the patient’s hand and rehabilitation robot in real-time. Then, the software system compares the obtained information with the expected values, and controls the rehabilitation robot to use the appropriate speed and angular speed to complete the required motion. When the system detects that the status exceeds the pre-specified threshold, it will automatically and slowly return to the safe state and inform the medical personnel in a timely manner. The Robot Research Center of Zhejiang University [50] designed a hand function rehabilitation training system that uses bending angle sensors, pressure sensors, and inertial sensors simultaneously. This system can identify the patient’s hand actions, spatial motion state, and pressure value of each part during grasping in real-time, and map them to the virtual hand. It is combined with virtual reality to motivate patients’ training through more exciting games.

#### 2.2.3. Optical Sensor-Based Hand Function Rehabilitation Robot

Conventional sensors are usually directly mounted on hand rehabilitation robots, which are difficult to integrate with actuators, complex in structures, and challenging to install on flexible actuators. Sensors made of elastomers or conductive liquids will deform when subjected to surface pressure, resulting in non-linear output and compromising accuracy. Optical sensors applied to hand rehabilitation robots mainly include optical fiber sensors and photoelectric sensors. The fiber optic sensor has high sensitivity, a compact structure, and strong anti-interference ability, which means it cannot affect the hand motion when applied to the hand rehabilitation robot. Optical sensors generally work by measuring the deviation of a light beam incident on a photosensitive surface.

He et al. [51] proposed an optical fiber pressure sensor for the hand rehabilitation exoskeleton. This sensor consists of a small piece of optical fiber package attached to a rigid 3D-printed structure. It can capture the interaction force on the exoskeleton in real-time to obtain the current hand motion and grasping situation. The sensor is small, simple, sensitive, safe, and low-cost, and can be easily integrated into the exoskeleton of the hand without affecting actuation. In a later study, the team also proposed a low-cost micro-hand posture sensor based on photoelectric technology that can be integrated into the hand exoskeleton [52], which measures the posture of multi-segment continuous structures in the hand rehabilitation exoskeleton. The sensor has low energy consumption, low noise, high sensitivity, and good real-time performance. It can be used to accurately control the rehabilitation robot, assess patient hand function, and adjust the rehabilitation plan.

Diez et al. [53] proposed a lightweight hand exoskeleton with micro-optical force sensors, which also applies optical principles and can measure human–machine interaction forces to estimate user intentions in rehabilitation scenarios. Due to its miniaturization, the sensor can be inserted between the human interface and the force transmission element. When the patient tries to act, the device can measure the force between the exoskeleton and the patient, identify the patient’s intention, and continuously control the exoskeleton to move in the desired direction to achieve the target posture. When the patient has a particular hand motor function and the force measured by the sensor exceeds a pre-calibrated threshold, the exoskeleton will automatically start a complete movement.

Liu Chenglong et al. [54] from Huazhong University of Science and Technology developed a soft actuator for the hand rehabilitation robot. It is embedded with optical fiber curvature sensors. When the finger is bent, the sensor senses different surface pressures and outputs voltage signals in a linear relationship. The software part can convert voltage signals into angle information, and then recognize the gesture. In this way, the rehabilitation robot can be tracked and controlled more accurately to assist in each training mode. At the same time, the system can also display the current gesture with 3D texture animation. The PMMA material is used as the light guiding medium of the sensor, which can effectively eliminate the interference of the extrusion deformation caused by the soft actuator’s inflation on the sensor’s output and make the output more stable. Table 1 presents a summary of the advantages and disadvantages of these hardware devices for functional hand rehabilitation systems included in the review.

## 3. Artificial Intelligence Used in Hand Rehabilitation Robots

In addition to the above-mentioned research on hardware devices for hand movement recognition, hand rehabilitation robots also require research in the field of software, combining hardware and software to achieve intelligence and wisdom in the devices. With the development of artificial intelligence technology, many new fields and products have emerged. Artificial intelligence can be applied to various modules of hand rehabilitation robots, such as hand movement recognition, control of hand rehabilitation robots, human-machine intelligent collaboration, interactive game design, training program design and result evaluation, cloud platform, structural design and optimization of hand rehabilitation robots, and so on. However, in practice, the research and application of artificial intelligence in hand rehabilitation robots are still not significant or not deep enough due to technical limitations and insufficient deployment of related medical facilities [7].

### 3.1. Gesture Recognition Algorithm

In hand rehabilitation robots, gesture recognition mainly requires real-time isolated dynamic gesture recognition, and artificial intelligence algorithms can be used for this recognition. The core idea is mainly to build a matching model from the data training set and then use the model to predict the recognized gestures. The more commonly used methods are mainly linear discriminant analysis (LDA), support vector machines (SVM), convolutional neural networks (CNN), long short-term memory (LSTM) and recurrent neural networks (RNN). For example, Zhang Guangxing [55] from Qingdao University of Science and Technology designed an integrated wrist rehabilitation robot using the LDA method that can recognize five kinds of movements, with an accuracy rate of over 90%. Zhang Fahui [56] from Nanchang University used the SVM model to recognize four kinds of gestures, with an average accuracy rate of 99.3%. Liu Wei [35] from Nanjing University of Aeronautics and Astronautics developed a hand exoskeleton rehabilitation robot using the convolutional neural network, which can recognize four kinds of gestures, with an average accuracy rate of 96.18%. Zhang Jianxi [57] designed a hand rehabilitation robot using a combination of RNN and LSTM algorithms for the recognition of nine gestures, with an average accuracy of 91.44%.

### 3.2. Control of the Hand Rehabilitation Robot

Safety issues need to be addressed in the control strategy or algorithm of the hand rehabilitation robot. More precise control strategies are needed, as well as balancing the ratio between the degree of control the robot has over the hand and the risk of miscalculation. For example, Jun Wu [58] designed a pneumatic flexible hand rehabilitation robot, used a sliding mode control algorithm based on fuzzy compensation to control the pneumatic muscles, proposed a dynamic surface control based on a nonlinear interference observer to realize the control of the pneumatic muscle system, and used an echo state network (ESN) with recursive least squares (RLS) for PID parallel adaptive control. Yihao Du et al. [59] proposed an adaptive control strategy based on a variable impedance equation model, which can combine the desired trajectory identified by physiological signals to obtain the final trajectory and calculate the required motion of each joint of the rehabilitation robot. Yonghao Yin [60] from Yanshan University also used RBF neural networks to approximate the compensation of errors caused by external perturbations and uncertainties to control the rehabilitation robot to achieve the desired results.

### 3.3. Human-Robot Intelligence Collaboration

Artificial intelligence can also be applied to the human-robot collaboration of hand rehabilitation robots, mainly in the assisted training mode. For example, Wang Xiangyu [61] from Harbin Institute of Technology designed an impedance control system to carry out motion following the hand rehabilitation robot in the assisted mode, which can identify the bending angle and muscle strength of the patient’s fingers when the patient’s hand has a certain behavioral ability and apply the appropriate force to help the patient complete the movement, and the system uses a fuzzy neural network.

### 3.4. Interactive Game Design

In order to increase patients’ interest and motivation during rehabilitation training, the hand rehabilitation robot can be equipped with some interactive games, such as interactive control based on voice recognition, virtual reality games based on visual recognition, and games based on brain–computer interface, etc. Artificial intelligence is also applied here. For example, Zhu Xikun [62] from Zhengzhou University designed a finger rehabilitation training system that can interact with active modes through both gesture recognition and voice recognition using algorithms such as neural networks and hidden Markov models. Cao Yali [63] designed a hand function rehabilitation robot software system that uses Unity3D to design different virtual games for different training modalities. Mou Yang et al. [64] designed an Android-based portable virtual reality rehabilitation device that can be applied to a hand rehabilitation robot. Ying Zhang [65] from Beijing University of Posts and Telecommunications designed a monocular vision-based hand grasping interaction training module.

### 3.5. Training Program Design and Outcome Evaluation

Traditional rehabilitation training programs and assessment of rehabilitation results are mostly conducted by doctors themselves, but this requires a lot of medical resources and sometimes has a certain subjective component. Theoretically, the application of artificial intelligence can improve this problem by digitizing and standardizing the assessment report and training program, which can identify the patient’s current hand movement status and physiological data in real time, combine it with the relevant medical program data, assess the patient’s current muscle strength level and rehabilitation training effect, and intelligently design the training program based on the assessment results. However, the actual effectiveness of these applications has yet to be clinically validated. For example, Neofect [66] has developed a Rapael Smart Glove, which has an intelligent system that can provide segmented rehabilitation recommendations and evaluation reports, and its unique RAPAEL intelligent rehabilitation algorithm that can build games that meet the patient’s rehabilitation plan.

### 3.6. Cloud Platform

The cloud platform can be used for one-stop implementation of training program design and evaluation of training results, as well as remote rehabilitation and real-time interaction. The cloud platform can monitor equipment parameter data in real time and provide safe and reliable rehabilitation training for patients. It can also apply digital twin technology to break the limitation of time and space and share big data sets of patients’ medical records, intelligent rehabilitation assessment reports, and training reports. Through these data, training programs are intelligently designed and pushed to rehabilitation robots, with interactive games, real-time remote training, and full cloud health management. For example, Xinyu Tang [67] from Southeast University designed a rehabilitation training and assessment system that uses cloud platform technology to aggregate, store, analyze, and display patient rehabilitation data. The system uses the DTW algorithm to compare the similarity of patient movement data streams captured by Kinect with standard movement data streams and uses them as indicators to assess the rehabilitation status of patients.

### 3.7. Hand Rehabilitation Robot Structure Design and Optimization

In addition to the software applications mentioned above, artificial intelligence can be applied to the hardware structure of the rehabilitation robot. For example, AI can optimize the mechanical mechanism based on human body data to make the machinery more ergonomic and better fit the biological curve of the human body to enhance the use experience.

## 4. Problems and Prospects of the Current Study

The previous context introduces the hardware and software of existing research on hand function rehabilitation robots. However, problems also exist that require further attention in future research. The following list summarizes the existing problems and limitations of the employed techniques in the current studies.

(1)Hand function rehabilitation equipment based on computer vision techniques is sometimes cumbersome, requires affixing markers, and is not convenient or portable enough for the external environment. Some aspects of the equipment will even make users feel dizzy. When using monocular vision, the recognition accuracy is affected by the light and color of the environment, skin color, etc. Although these problems can be improved to some extent when using binocular vision, it requires more complicated calibration and correction, which increases the computational cost and creates other interference factors that affect the image quality. Therefore, it is necessary to continue to improve the equipment and optimize the algorithm.(2)When using computer vision technology for bare hand gesture recognition and training patients’ hand function with games in the virtual environment, the tactile feedback of the affected hand is often not available, ignoring the motion angle and time that needs to be strictly set according to the patient’s motion ability to ensure the safety of the game. The feedback is mandatory during sensorimotor rehabilitation. Therefore, more attention should be paid to the design by adding some threshold limits.(3)When using computer vision technology only for gesture recognition and rehabilitation training, the type of data is relatively simple. When recognizing complex gestures, it is inevitable to encounter a situation where the acquisition cannot be collected or where the acquisition and recognition are inaccurate. Some other information collection methods, such as physiological signals and so on, can be used to achieve multi-information fusion.(4)Most hand rehabilitation robots based on wearable device gesture recognition need to wear additional sensors, which increases the cost and at the same time is inconvenient to use and has certain restrictions for patients. The recognition algorithm and the central processor of the wearable device have higher requirements, so there may be some delays. The algorithm also needs to be retrained if the number of sensor positions changes. Therefore, the balance and optimization between comfort, diversity, stability, accuracy, and timeliness in acquisition need to be further studied, and the safety of unsupervised training also deserves attention.(5)The hand rehabilitation robots based on physiological signal gesture recognition are prone to instability in signal acquisition and transmission, and the amount of data that can be recognized is limited, making it difficult in the application to fine finger motion rehabilitation and the recognition of continuous motion. To address this aspect, multi-mode control strategies can also be used, in conjunction with kinematic signals, optical signals, etc.(6)Regardless of the techniques used, the rehabilitation robot should be designed with a closed-loop control network, which can, for example, provide force feedback, angular feedback, position feedback, haptic feedback, etc. through sensors to provide timely feedback and adjust parameters according to the real-time status of patients.(7)Most hand function rehabilitation robots are still in the research stage, and those based on computer vision technologies require far more attention than those found on wearable sensors. Evaluated using the “Technology Readiness Level (TRL) system” by Mankins, most hand function rehabilitation systems based on computer vision technology are still at TRL5 or 6 (technology demonstration). A few of them are at TRL7 or 8 (system/subsystem development), and few reach TRL9 (system test, deployment and ops, the highest level) [68]. On the other hand, those based on wearable devices are at least TRL5 to 7, and many of them reach TRL9.

Besides the above existing problems, the clinical utility of hand function rehabilitation robots has been evaluated in terms of appropriateness, acceptability, and practicability [69]. Most established rehabilitation robots had effective appropriateness, while some robots still in the research stage had relevant appropriateness. Participants’ acceptability of rehabilitation robots and computer-vision-based games is mainly related to their effectiveness. Some said that the games could increase their interest in training, and some expressed concerns, such as comfort. Speaking of practicability, most of them perform well in terms of functionality and suitability, but some technical problems hinder the user experience. The variety of functionality and wearing comfort still require further improvements.

In the future, we expect that a hand function rehabilitation robot should first be able to fully simulate a human therapist’s approach and achieve complete “human-machine integration”. Secondly, it can be upgraded under human guidance to assess the patient’s condition, give rehabilitation training plans and evaluate the training effect. Moreover, it can realize a more convenient and intelligent human–machine interaction mode. Finally, it can also assist in optimizing the allocation of medical resources to achieve efficient and interesting telemedicine.

## Figures and Tables

**Table 1 brainsci-12-01079-t001:** This table is a summary of the different input device for the hand function rehabilitation system and indicates the main advantages and disadvantages of these technologies.

Input Device	Specific Device	Reference	Advantage	Disadvantage
Camera	Virtual game	[17,18,19,20,21,22,23,24]	Non-invasive; does not require wearing extra equipment; easy to use	Has high requirements for the external environment; low recognition speed and the accuracy rate; may need markers
	Robot	[25,26,27,28,29,30,31,32,33]		
Wearable device	Physiological signal sensor	[34,35,36,37,38,39,40,41]	Can recognize more gestures with small input data and high precision in real time; has good robustness	More expensive; require wearing extra equipment; easy to cause fatigue; requires calibration
	Kinematics sensor	[42,43,44,45,46,47,48,49,50]		
	Optical sensor	[51,52,53,54]		

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
