# Peer review of "A Review of Hand Function Rehabilitation Systems Based on Hand Motion Recognition Devices and Artificial Intelligence"

_brainsci, 2022, doi:10.3390/brainsci12081079_

Round 1
Reviewer 1 Report
Dear authors,
It was a pleasure to revise your paper. I believe that it needs minor corrections, namely, as you can read in the version that I am sending in attachment.
Best regards,

Author Response
Dear reviewer,
We appreciate you very much for your positive and constructive comments and suggestions on our manuscript. (ID:brainsci-1833897).
We have studied your comments carefully and have made revision. We have tried our best to revise our manuscript according to the comments. Please see the attachment.
We would like to express our great appreciation to you and reviewers for comments on our paper. Looking forward to hearing from you.

Reviewer 2 Report
Thank you for the opportunity to read and review this interesting and original article.
Its main strength is the originality, since no other revisions on the subject have been carried out.
However, there are many concerns that must be assessed to improve the review.
Abstract
Please consider elimination of the sentence l. 17-19 (“The recovery… century), or moving it after the next sentence. It doesn´t really fit where it currently is.
Introduction
l. 35. I think “motor function” would be a better option than “ROM” because the limitations are not only secondary to reduced ROM, but are also caused by poor strength, poor coordination, spasticity…
Please review the second paragraph since the three principles you mention are not clear. Also, references are needed to support this paragraph.
Third paragraph needs references.
Main body
The information is too vast and confusing. Please summarize and prioritize it.
You could maybe summarize the information in Tables focusing on advantages and disadvantages of each technology.
It would be interesting to evaluate the clinical utility of each technology. You can refer to the following paper: Spa B. A multi-dimensional model of clinical utility. 2006. International Journal for Quality in Health Care; 18(5): 377-382.
Also, since you mention that the majority of the systems are in the laboratory research stages, you could mention the level of each system in terms of the technology readiness level (Mankins JC. Technology readiness assessments: a retrospective. 2009. Acta Astronautica).
l. 90. What do you mean by “cognitive abilities in the body”?
l. 407. Please consider mentioning every method, instead of “and so on”.
l. 459. “by standardizing and standardizing”, is it correct?
Author Response

(The authors gave the same response as above.)

Round 2
Reviewer 1 Report
Thank you for this updated version. I have nothing more to suggest/amend.
Reviewer 2 Report
Thank you for this updated version. I have nothing else to suggest.